

# Comparison of bacterial diversity and abundance between sexes of *Leptocybe invasa* Fisher & La Salle (Hymenoptera: Eulophidae) from China

Chunhui Guo[1], Xin Peng[1], Xialin Zheng[2], Xiaoyun Wang[2], Ruirui Wang[1], Zongyou Huang[2] and Zhende Yang[1,3]

[1] College of Forestry, Guangxi University, Nanning, Guangxi, China
[2] Guangxi Key Laboratory of Agric-Environment and Agric-Products Safety, College of Agriculture, Guangxi University, Nanning, Guangxi, China
[3] Guangxi Key Laboratory of Forest Ecology and Conservation, College of Forestry, Guangxi University, Nanning, Guangxi, China

## ABSTRACT

**Background**. Insects harbor a myriad of microorganisms, many of which can affect the sex ratio and manipulate the reproduction of the host. *Leptocybe invasa* is an invasive pest that causes serious damage to eucalyptus plantations, and the thelytokous parthenogenesis, low temperature resistance, protection in galls, generation overlap and small body of *L. invasa* contribute to its rapid invasion and population growth. However, the endosymbiotic bacterial composition, abundance and sex differences of *L. invasa* remain unclear. Therefore, this research aimed to identify the bacterial communities in *L. invasa* adults and compare them between the sexes of *L. invasa* lineage B.

**Results**. The Illumina MiSeq platform was used to compare bacterial community composition between females and males of *L. invasa* by sequencing the V3–V4 region of the 16S ribosomal RNA gene. A total of 1,320 operational taxonomic units (OTUs) were obtained. These OTUs were subdivided into 24 phyla, 71 classes, 130 orders, 245 families and 501 genera. At the genus level, the dominant bacteria in females and males were *Rickettsia* and *Rhizobium*, respectively.

**Conclusion**. The endosymbiotic bacteria of *L. invasa* females and males were highly diverse. There were differences in the bacterial community of *L. invasa* between sexes, and the bacterial diversity in male specimens was greater than that in female specimens. This study presents a comprehensive comparison of bacterial communities in *L. invasa* and these data will provide an overall view of the bacterial community in both sexes of *L. invasa* with special attention on sex-related bacteria.

## INTRODUCTION

There are numerous microorganisms living in insects, including bacteria, fungi, yeast and viruses, that play a vital role in the growth and reproduction of host insects (*Dillon & Dillon, 2004*; *Doğanlar, 2005*; *Crotti et al., 2012*; *Frago, Dicke & Godfray, 2012*; *Engel*

Corresponding author
Zhende Yang, dzyang68@126.com

& *Moran, 2013*; *Hammer & Bowers, 2015*). Over the course of long-term coevolution, microorganisms develop a close relationship with host insects, which may have an effect on the reproduction, survival, community interactions, and the ability to resist predators and vectors of the hosts (*Oliver et al., 2003*; *Oliver et al., 2010*; *Moran, 2007*; *Clark et al., 2008*; *Moran, McCutcheon & Nakabachi, 2008*; *Moya et al., 2008*). In light of the significant functions of the microorganisms, they have received much attention from the international academic community. In some insects, the diversity and function of endosymbiotic bacteria have been well studied. For instance, the bacteria in termites are mainly *Bacteroidetes, Firmicutes* and *Actinobacteria* and can assist their hosts in breaking down lignocellulose and promoting the nitrogen cycle (*Warnecke et al., 2007*; *Brune, 2014*). The bacteria in *Aphis gossypii* improve its resistance and adaptation (*Łukasik et al., 2013a*; *Łukasik et al., 2013b*). In recent years, manipulating endosymbionts for pest control has raised wide concern, and its theory and methods have been applied successfully to some extent. Introduction of antimalarial endosymbionts into the mid gut of host pests could inhibit the breeding of plasmodia and in turn reduce the efficiency of mosquito transmission of malaria (*Wang et al., 2012*). Mixed application of antibiotics and insecticides effectively reduced the quantity of endosymbionts in *Nilaparvata lugens* while improving the control effect of insecticides (*Shentu et al., 2016*). Based on research on the related incompatible insect technique (IIT), researchers used the maternally inherited endosymbiotic bacterium *Wolbachia* for sterilization, which had good effects on eliminating the fecundity of mosquitoes (*Zheng et al., 2019*). Obviously, it is necessary to clarify the bacterial composition and diversity in insects, which are the bases of manipulating endosymbionts for pest control. In addition, previous investigations have shown that sex is an important factor affecting bacterial diversity. For example, due to different attack behaviors, the overall diversity and richness of bacterial communities associated with female *Dendroctonus valens* are higher than those associated with males of this beetle species (*Xu et al., 2016*). The bacterial composition of mosquitoes was also affected by sex (*Minard, Mavingui & Moro, 2013*; *Zouache et al., 2011*). Different anatomies and life histories between male and female flies could provide differential opportunities for bacterial colonization (*Tang et al., 2012*).

The blue gum chalcid *Leptocybe invasa* Fisher & LaSalle (Hymenoptera: Eulophidae: Tetrastichinae) is a cosmopolitan pest that damages many *Eucalyptus* species (*Mendel et al., 2004*; *Le et al., 2018*). *L. invasa*, originating in Australia, was first recorded in 2000 and has since been discovered in 45 countries of Asia, Europe, Africa, Oceania and America (*Le et al., 2018*; *Zheng et al., 2014a*). A new study demonstrated that an increasing number of areas will become suitable for *L. invasa* due to climate warming (*Huang et al., 2019*). Every delicate twig, vein and petiole of eucalyptus trees may provide a spawning ground for this pest, and galls ultimately lead to stunted growth of the trees, causing great losses in local eucalyptus plantations (*Mendel et al., 2004*; *Zheng et al., 2014a*; *Huang et al., 2018*). DNA barcode data indicated that *L. invasa* includes two genetically separate lineages (lineages A and B). Researchers considered the Italian, Argentinean and Tunisian populations to belong to lineage A and the Chinese population to belong to lineage B (*Le et al., 2018*; *Dittrich-Schröder et al., 2018*). The absence of natural enemies, presence of large amounts of suitable host plants, small size, protection in galls, strong resistance to low temperature

and thelytokous parthenogenesis of *L. invasa* caused its rapid invasion and growth of in China (*Zheng et al., 2014a*). As a result, it has become one of the most difficult pests to control (*Zheng et al., 2014a*; *Huang et al., 2018*; *Le et al., 2018*). It is important in tems of theory and application to study the endosymbiotic bacterial diversity of *L. invasa* and then control the wasps by using these endosymbiotic bacteria.

To date, few studies have reported on the overall endosymbiotic bacteria of *L. invasa*, which is an invasive gall-inducing insect. Only a few studies have comprehensively examined the endosymbiotic bacteria in this species. *Wang et al. (2018)* cultured 11 strains from female adults of *L. invasa* in winter using traditional methods and classified them into three phyla (*Firmicutes, Actinobacteria*, and *Proteobacteria*), three classes (*Bacilli, Actinobacteria,* and *Gammaproteobacteria*) and four orders (*Bacillales, Micrococcales, Lactobacillales,* and *Enterobacterales*) that were related to growth, development, nutrition metabolism and immunity. *Nugnes et al. (2015)* researched the bacteria living in adults among different populationsvia denaturing gradient gel electrophoresis (DGGE) analysis and found that *Rickettsia* occurred in the reproductive tissues of female *L. invasa*, suggesting a relationship with its thelytokous parthenogenesis. *L. invasa* harbors a myriad of bacteria, and bacterial differences between sexes have strong effects on insects, such as effects on reproductive regulation (*Wang et al., 2018*; *Nugnes et al., 2015*). Therefore, the overall endosymbiotic bacterial composition and abundance of *L. invasa* and the differences between sexes are important to study.

In this study, the endosymbiotic bacteria in female and male adults of *L. invasa* were indentified by 16S rRNA sequencing of the V3–V4 region to shed light on their internal bacterial compositions. The females and males were also compared to address sexual differences in the endosymbionts. These results will provide a valuable bacterial pool for *L. invasa* and will further contribute to understanding its reproductive strategies and invasion mechanisms.

## MATERIALS & METHODS

### Insect sampling

Branches of DH 201-2 (*Eucalyptus grandis* × *E. tereticornis*) (Myrtales: Myrtaceae) harboring galls of *L. invasa* were removed from the Teaching and Experiment Base of Forestry College, Guangxi University (108°17′E, 22°51′N), Nanning city, Guangxi Zhuang Autonomous Region from July to August 2018. The branches were placed in a plastic bottle filled with water to retain freshness and transferred into a sealed net cage (40 cm × 40 cm × 80 cm) at room temperature to keep the adults from escaping. The water in the plastic bottle was renewed daily until the emergence of *L. invasa* adults. Sexes were identified by morphological observation (*Zheng et al., 2014b*).

### DNA extraction

Fifty adults of each sex of *L. invasa* newly emerged within 12 h were fasted for 6 h. Then, both samples were sterilized externally with 75% ethanol for 2–5 min and rinsed 3 times with sterilized water to remove microbes on the surface. The total bacterial DNA of each sample was extracted using a Power Soil DNA Isolation Kit (MO BIO Laboratories)

according to the manufacturer's instructions. The quality and quantity of DNA were assessed by the ratios of 260 nm/280 nm and 260 nm/230 nm. Then, the qualified DNA was stored at −80 °C for further processing. The DNA of each individual was extracted by using a Chelex-100 and proteinase K-based method (*Gebiola et al., 2009*).

## PCR amplification and cloning of the bacterial 16S rRNA gene

Amplification of the V3–V4 hypervariable region of the bacterial 16S rRNA gene was performed by using the universal bacterial primers 338F (5′-ACTCCTACGGGAGGCAGCA-3′) and 806R (5′-GACTACHVGGGTWTCTAAT-3′). PCRs were carried out in 50 μL solutions containing 10 μL of 10 × buffer, 0.2 μL of Q5 High-Fidelity DNA Polymerase, 10 μL of High GC Enhancer, 1 μL of dNTPs, 10 μM each forward and reverse primer, 60 ng of genomic DNA and enough ddH$_2$O to reach 50 μL. The amplifications were performed in a ABI Applied Biosystems 9902 thermal cycler with an initial denaturation step at 95 °C for 5 min, followed by 35 cycles of annealing and extension (each cycle consisted of 95 °C for 1 min, 50 °C for 1 min and extension at 72 °C for 1 min) and a final extension at 72 °C for 7 min. The PCR products were checked by electrophoresis on an agarose gel (1.8% agarose, 1 × TBE), stained with ethidium bromide and visualized under ultraviolet light. The products from the first round of PCR were purified with VAHTS$^{TM}$ M DNA Clean Beads. The second round of PCR was then performed in a 40 μL reaction containing 20 μL of 2 × Phμsion HF MM, 8 μL of ddH$_2$O, 10 μM each forward and reverse primer and 10 μL of PCR product produced in the first round. The second round of PCR was run under the following conditions: initial denaturation at 98 °C for 30 s, followed by 10 cycles at 98 °C for 10 s, 65 °C for 30 s and 72 °C for 30 s and a final extension at 72 °C for 5 min. Finally, all PCR products were quantified and pooled by Quant-iT$^{TM}$ dsDNA HS Reagent. High-throughput sequencing analysis of bacterial rRNA genes was performed on the purified, pooled sample by using the Illumina HiSeq 2500 platform at Biomarker Technologies Co., Ltd., Beijing, China.

## Bioinformatics and statistical analysis

After sequencing, PE reads obtained with HiSeq sequencing were merged by overlapping to obtain raw tags. To obtain clean tags, the raw tags were denoised, sorted and separated by using Trimmomatic (version 0.33). The remaining sequences were filtered for redundancy, and all unique sequences in each sample were clustered into operational taxonomic units (OTUs) on the basis of 97% similarity. Low-abundance OTUs were identified and eliminated by using UCHIME v4.2. Taxonomic assignment of the OTUs was conducted with the Silva reference database. Species abundance tables were generated by QIIME, and community structures in every taxon category was plotted by R software. The relative abundances of the bacteria were determined by percentages.

Alpha diversity based on Chao1 richness and ACE richness estimators, as well as the Simpson and Shannon diversity indices, was evaluated by using the mothur v.1.11.0 program. Among these measure, Chao1 and ACE reflected species richness in the samples, the Shannon index reflected community diversity, the Simpson index reflected the dominance of species in the community, and the coverage index reflected the degree

to which the sequencing results represented the actual composition of the microorganisms in the samples.

## Molecular characterization and phylogenetic analyses

COI was amplified by using the forward primer LCO1490 and reverse primer HCO2198 (*Nugnes et al., 2015*). The 16S rRNA gene of *Rickettsia* was amplified by using the primers listed in Table S1. The PCR program for both genes (COI and 16S rRNA) was as follows: 3 min of initial denaturation at 94 °C, 30 cycles at 94 °C for 30 s, 55 °C for 30 s and 72 °C for 1 min and a final extension of 5 min at 72 °C. PCR products were observed by using 1.0% agarose gel electrophoresis, and the amplified fragments were directly sequenced by TsingKe Biological Technology Co., Ltd, Beijing, China. Representative sequences of other regions were downloaded from GenBank, and sequence alignment was completed by using Clustal X. The neighbour-joining method was used to construct a consensus phylogenetic tree with MEGA7 software. To evaluate the branch support of the phylogenetic tree, bootstrap analysis of 1,000 replicates was performed.

## Accession numbers

Data is available at NCBI SRA, accession numbers: SRR9591039, SRR9591038. The COI and 16S rRNA sequences determined in this study have been deposited in the GenBank database with Accession number MN524231 and MN524230, respectively.

# RESULTS

## Sex of *L. invasa* specimens in this study

All female and male specimens were identified on the basis of morphology. In this study, a total of 656 females and 51 males were collected (Table S2). The materials were deposited at the Forest Conservation Laboratory, College of Forestry, Guangxi University, Nanning 530004, China.

## Sequencing and classification

A total of 533,266 raw tags (370,680 from males and 162,586 from females) were obtained for *L. invasa,* and 476,235 clean tags (328,833 from males and 147,402 from females) were generated (Table S3), which were classified into different OTUs based on 97% similarity. Among the 476,235 clean tags, a total of 1,320 OTUs were obtained; of these 1,320 OTUs, 154 were common to both sexes, and 38 and 1128 were specific to female and male adults, respectively (Fig. 1).

## Analysis of alpha diversity

Alpha diversity was estimated by five indices: Chao1, the Shannon index, the Simpson index, the ACE and coverage. The results in Table 1 show that the bacteria in *L. invasa* adults were diverse in both sexes. The Chao1 (229.50 vs 1282.00) and ACE (212.84 vs 1282.28) values were lower in the females than in the males. Good agreement was also observed between the Simpson and Shannon indices. The Shannon index (0.59 vs 6.13) was lower in the females than in the males, while the Simpson index (0.85 vs 0.01) was higher in the female wasps than in the male wasps, indicating that the diversity of the bacterial
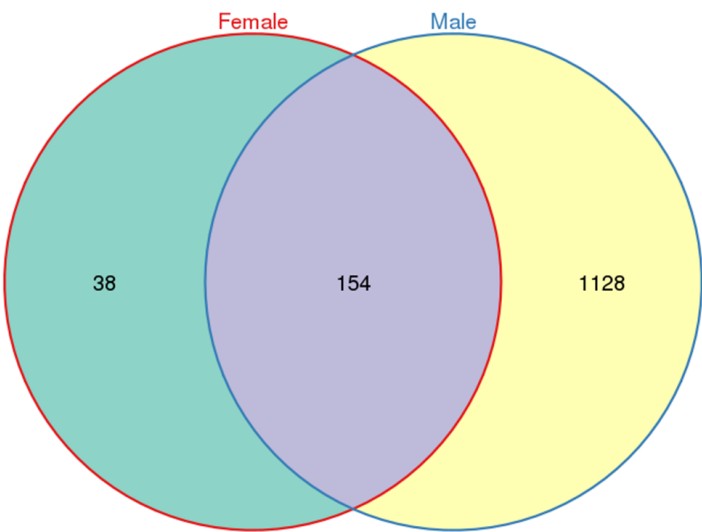

**Figure 1** **Venn diagram of OTU distribution in *Leptocybe invasa* female and male adults.** Numbers within compartments indicate OTU counts of according to mathematical sets.

**Table 1** **Statistics of alpha diversity indices of the bacteria in female and male adults of *Leptocybe invasa*.**

| Sample | ACE | Chao1 | Simpson | Shannon | Coverage |
|--------|-----|-------|---------|---------|----------|
| Female | 212.84 | 229.50 | 0.85 | 0.59 | 1.00 |
| Male | 1282.28 | 1282.00 | 0.01 | 6.13 | 1.00 |

community in males was higher than that in females. The coverage was near 100% for both males and females, illustrating a higher probability of bacteria being detected than of bacteria being undetected.

## The analysis of community composition and species abundance

The bacterial community composition and species abundance in both sexes of *L. invasa* were analyzed (abundances greater than 0.1%) based on the results of the OTUs (Table 2, Fig. 2). A total of 24 phyla were detected and classified in the samples. *Proteobacteria* was the dominant bacterial phylum annotated in females and males, accounting for 95.63% and 34.99% of bacteria, respectively. At the genus level, *Rickettsia* (with an abundance of 93.67%) and *Rhizobium* (with an abundance of 5.73%) were the dominant bacteria in females and males, respectively. In addition, it was noteworthy that the abundance of *Rickettsia* was less than 1% in males (Table 3).

## Molecular characterization and phylogenetic analyses

After comparison with Genebank, the identification of *L.invasa* in this research was lineage B and the phylogenetic tree of COI also indicated that the population of this research

**Table 2  Basic composition of the bacterial colonies in female and male adults of *Leptocybe invasa*.**

| Sample | Phylum | Class | Order | Family | Genus |
|---|---|---|---|---|---|
| Female | 10 | 26 | 44 | 76 | 122 |
| Male | 24 | 69 | 127 | 238 | 487 |
| Female-specific | 0 | 2 | 3 | 7 | 14 |
| Male-specific | 14 | 45 | 86 | 169 | 379 |
| Sex-in common | 10 | 24 | 41 | 69 | 108 |
| Total | 24 | 71 | 130 | 245 | 501 |

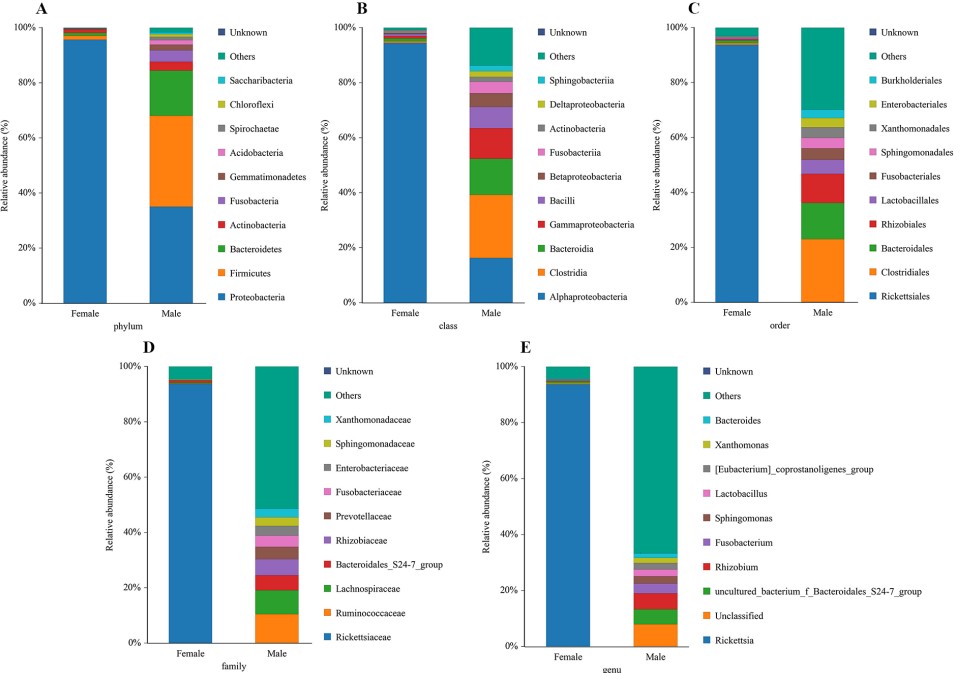

**Figure 2  Relative abundance of top 10 bacteria at the levels of phylum (A), class (B), order (C), family (D) and genu (E) in females and males of *Leptocybe invasa*.**

belonged to the lineage (Fig. 3). The phylogenetic analysis of 16S rRNA genes revealed that the *Rickettsia* of *L. invasa* symbionts belonged to the *Rickettsia* transitional group (Fig. 4).

# DISCUSSION

## Differences in bacteria between female and male adults

This research revealed that the bacteria harbored in *L. invasa* had high diversity, and many of the endosymbiotic bacteria were annotated in this species for the first time. Based on alpha diversity analysis, the diversity of the endosymbiotic bacteria in males was higher than that in females (Table 1). The variation in bacterial communities between males and females may be partly explained by differences in physiology structure and between the two sexes of *L. invasa*; specifically, the female wasps have ovaries, which harbor an abundance of *Rickettsia*, allowing the genus to occupy different bacterial niches than in

Table 3  Relative abundance of dominate bacteria at the levels of genus in female and male adults of *Leptocybe invasa*.

| Genus | Female (%) | Male (%) |
|---|---|---|
| *Rickettsia* | 93.67 | 0.04 |
| Uncultured_bacterium_f_*Bacteroidales*_S24-7_group | 0.71 | 5.37 |
| *Lactobacillus* | 0.31 | 2.38 |
| *Sphingomonas* | 0.25 | 2.62 |
| *Bacteroides* | 0.11 | 1.65 |
| *Fusobacterium* | 0.04 | 3.49 |
| [Eubacterium]_*coprostanoligenes*_group | 0 | 2.34 |
| *Rhizobium* | 0 | 5.73 |
| Unknown | 0 | 0.01 |
| *Xanthomonas* | 0 | 1.83 |
| Others | 4.48 | 66.68 |
| Unclassified | 0.44 | 7.86 |

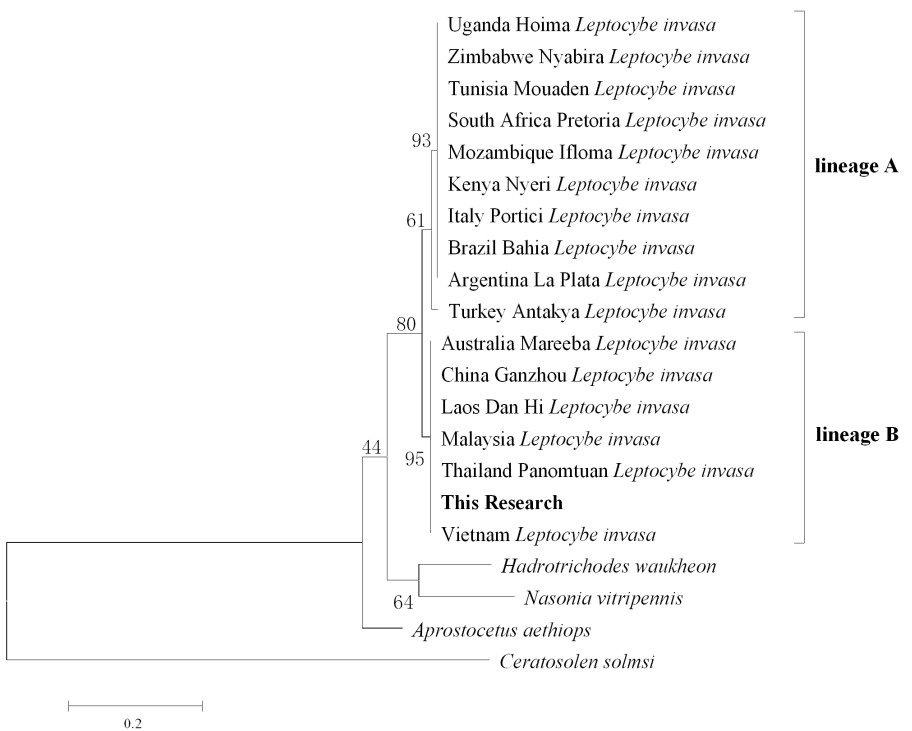

**Figure 3  Phylogenetic tree of different *Leptocybe invasa* populations based on COI sequences.**

males (*Nugnes et al., 2015*). Another possibility is that insects launch innate and systematic immune responses to cope with microbe colonization (*Leulier & Royet, 2009*) and females have stronger immune systems than males (*Kurtz et al., 2000*).

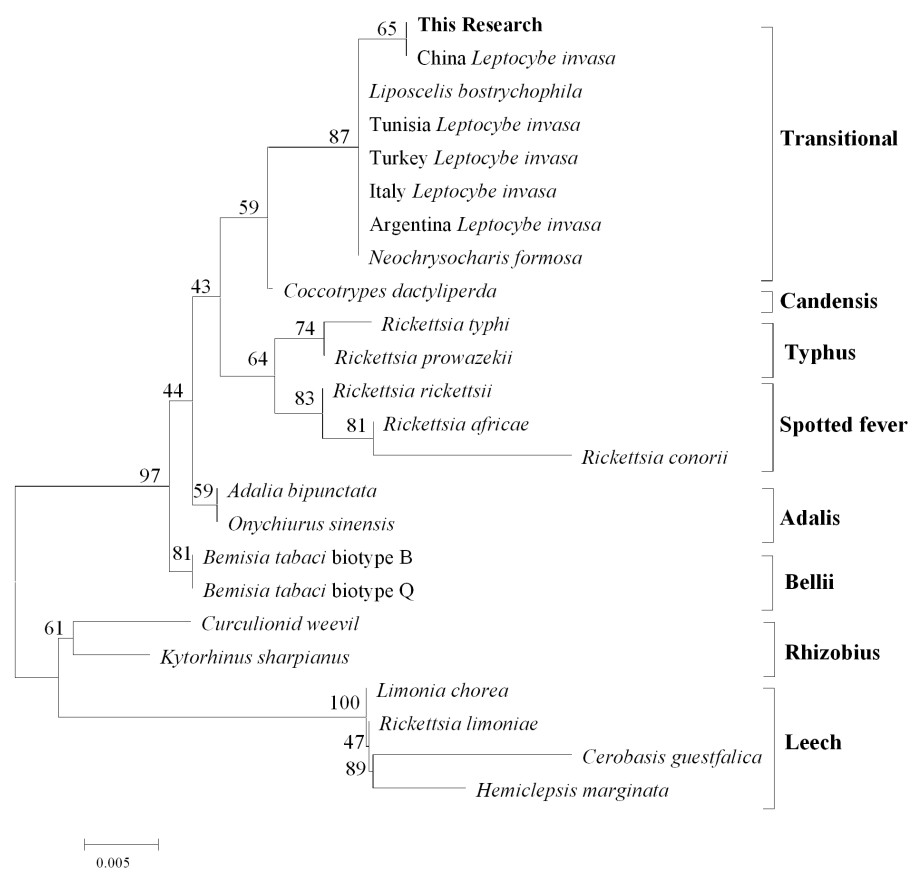

**Figure 4** Phylogenetic analysis of different *Rickettsia* groups based on their 16S rRNA sequences.

## Comparison of the bacteria with those in other insects

Bacterial community analysis at the phylum level demonstrated that *Proteobacteria* was the dominant group in female and male wasps, and *Firmicutes, Bacteroidetes, Actinobacteria* and *Fusobacteria* were also annotated. Previous studies revealed that *Proteobacteria* were dominant in other Hymenoptera, such as *Apis cerana* and leaf-cutter ants (*Ahn et al., 2012*; *Zhukova et al., 2017*). In contrast, *Firmicutes* and *Bacteroidetes* were the major bacterial phyla detected in the guts of termites (*Miyata et al., 2007*; *Xiang et al., 2012*) and bees (*Mohr & Tebbe, 2006*). *Firmicutes* and *Actinobacteria* were the dominant bacteria in *A. mellifera* and bumblebees (*Ahn et al., 2012*; *Praet et al., 2018*).

## Putative functions of dominant endosymbiotic bacteria in *L. invasa*

Several of the bacteria detected in this study are commonly described in insects at the genus level, and some have been found in Hymenoptera, such as honeybees (*Mohr & Tebbe, 2006*) and termites (*Xiang et al., 2012*). Intriguingly, two genera, *Staphylococcus* and *Escherichia*, are known to contain cultivable species (*Wang et al., 2018*). Gloverin and lysozyme gene expression was upregulated when silkworm larvae were fed *Escherichia* and *Staphylococcus*, indicating that the two bacteria were closely related to the immune signaling pathway of the silkworm (*Douglas, 2015*). We hypothesized that *Escherichia* and *Staphylococcus*

may also be involved in the immunoreaction of *L. invasa*. Functions have been suggested for some of the other bacterial genera detected in this study. The *Enterobacteriaceae* that are associated with insects help with digestion, the detoxification of toxic substances, and resistance to pathogens and enhance the adaptability of the host (*Anand et al., 2010*). Adding *Enterobacter* to feed extended the life span of Mediterranean flies (*Behar, Yuval & Jurkevitch, 2005*; *Behar, Yuval & Jurkevitch, 2008*). Similarly, *Enterobacteriaceae* (*Hongoh & Ishikawa, 2000*) and *Acinetobacter* (*Broderick et al., 2004*) facilitated carbon-nitrogen metabolism and accelerated the growth and development of host insects; e.g., the *Acinetobacter* belonging to termites have a nitrogen-transforming function according to *Warnecke et al.*'s (*2007*) research. Some bacteria associated with immunization were also discovered in *L. invasa*, such as *Lactobacillus*. *Lactobacillus* had some positive effects on insect resistance (*Xia et al., 2013*). In addition, *Bacillales* were also detected in this study and may be insect pathogens, such as *Bacillus thuringiensis* and *Bacillus cereus* (*Broderick et al., 2004*; *Raymond et al., 2010*; *Song et al., 2014*). In contrast, some *Bacillus* in termites might be involved in the degradation of cellulose and hemicellulose (*Konig, 2006*). In this study, *Bacillales* were detected in both sexes, and their specific functions require further study. Nevertheless, *Acinetobacter* was detected in *L. invasa,* and previous research showed that *Acinetobacter* produces an antiviral compound that inhibits tobacco mosaic virus (*Lee et al., 2009*). Moreover, members of *Bacteroidetes* specialize in the degradation of complex organic matter, including lignocellulosic compounds (*Yuki et al., 2015*). *Bacteroidetes* are also involved in the decomposition and metabolism of polysaccharides (*Xu et al., 2003*; *Sonnenburg et al., 2010*), which are beneficial for host absorption and digestion (*Liu et al., 2011*). In addition, *Bacteroidetes* also include some *Azotobacter*, such as *Azobacteroides pseudotrichonympha*, which can provide the host with amino acids for nutrition (*Noda et al., 2009*; *Desai & Brune, 2012*). *Bacteroidetes* involved in the degradation and fermentation of phytomass could influence the nutrient absorption of *L. invasa*, but further studies are needed. Many other groups of bacteria with undefined functions were detected in *L. invasa* for the first time in this study. Better knowledge of the bacteria associated with *L. invasa* will allow researchers to investigate their role in host biology.

A sequence similarity search revealed that *Rhizobium* was the dominant bacterium in male adults (Fig. 2, Table 3). *Rhizobium* produces a variety of enzymes with cellulose- and pectin-hydrolyzing activities that can hydrolyze the glycoside skeleton of the plant cell wall and play a very important role in the symbiosis between *Rhizobium* and leguminous plants (*Robledo et al., 2008*; *Huang et al., 2018*). *Rhizobium* is an endosymbiont detected in the gut of some phytophagous insects and can help the host synthesize nitrogen-containing substances that are lacking in food (*Russell et al., 2009*).

*Rickettsia* (with an abundance of 93.67%) was the dominant bacterial genus present in female adults (Fig. 2, Table 3). *Rickettsia* is a maternally inherited intracellular bacterium in a wide range of arthropods and is capable of controlling populations by reproductive manipulation, such as parthenogenesis inducing (PI) (*Hagimori et al., 2006*; *Adachi-Hagimori, Miura & Stouthamer, 2008*; *Giorgini et al., 2010*) and male killing (MK) (*Lawson et al., 2001*; *Von der Schulenburg et al., 2001*; *Majerus & Majerus, 2010*). During female gamete formation in *Rickettsia*-carrying *Neochrysocharis formosa*, meiotic cells

underwent only one equatorial division, and meiotic recombination was absent, which demonstrated that *Rickettsia* could induce parthenogenesis by changing the meiosis of wasps (*Adachi-Hagimori, Miura & Stouthamer, 2008*). *Rickettsia* also induced male embryo death in *Adalia bipunctata* and *A. decempunctata* (*Hurst, Majerus & Walker, 1993*; *Hurst, Walker & Majerus, 1996*; *Werren et al., 1994*). Moreover, *Rickettsia* affects the fitness of the host and protects it against adverse environmental conditions (*Oliver et al., 2003*; *Sakurai et al., 2005*; *Chiel et al., 2009*; *Himler et al., 2011*; *Brumin, Kontsedalov & Ghanim, 2011*). For instance, preadult development of the *Bemisia tabaci* B-biotype was faster with *Rickettsia* infection than without (*Chiel et al., 2009*). *Himler et al. (2011)* found that *Rickettsia*-carrying whiteflies produced more offspring, developed faster, had a higher rate of survival to adulthood, and produced a larger proportion of daughters than did uninfected whiteflies. Males have never been recorded in Italy, Tunisia and Argentina, and rarely in Turkey (sex ratio 0–0.5%) (*Nugnes et al., 2015*). These results show that *L. invasa* reproduces by thelytokous parthenogenesis. In contrast, males appeared more frequently in China, India and Thailand. In this study, the sex ratio was 7.2%. In addition, *Nugnes et al. (2015)* found that *Rickettsia* was located in reproductive tissues in females and passed to the next generation via vertical transmission, representing a possible reason for thelytokous parthenogenesis in *L. invasa*. Female *L. invasa* play an important role in invasion and colonization (*Zheng et al., 2014a*). The results of the current investigation could explain why the sex ratio in wasps is female-biased and support the hypothesis that *Rickettsia* can induce thelytokous parthenogenesis in *L. invasa*. However, both explanations require further testing. In addition, a low abundance of *Rickettsia* was present in males in this research. For Hymenoptera, the dominant reproductive mode is arrhenotoky; that is, diploid females develop from fertilized eggs, and haploid males develop from unfertilized eggs (*Van Wilgenburg, Driessen & Beukeboom, 2006*). A previous investigation suggested that *Rickettsia* could be passed to the offspring by vertical transmission (*Nugnes et al., 2015*), and a threshold density of *Rickettsia* bacteria in eggs is required to trigger the development of female embryos *Giorgini, 2001*; *Giorgini et al., 2010*. Removing *Rickettsia* by feeding antibiotics could lead to the production of more male offspring. *Giorgini et al. (2010)* found that *Rickettsia*-infected *Pnigalio soemius* generated only female progeny, and after 24 h, when the *Rickettsia* was removed by 20 mg/mL rifampin, adults produced almost all male offspring. *Hagimori et al. (2006)* declared that *Rickettsia* was related to the thelytokous parthenogenesis of *N. formosa*, a dominant parasite of leaf miners, and after removing *Rickettsia* from the adults by feeding the adults tetracycline, female offspring without *Rickettsia* were present. Therefore, future studies should clarify whether *Rickettsia* is involved in the reproductive manipulation of *L. invasa* accomplished via feeding with antibiotics. Furthermore, environmental factors could also influence the density of the bacteria, and endosymbiont densities and functions may change with space, time and season (*Bordenstein & Bordenstein, 2011*; *Nugnes et al., 2015*). A previous study indicated that the sex ratio of the Chinese population could change with temperature, presumably because the relationship between *Rickettsia* strain and Chinese population is weaker than that of Western population, which could be more susceptible to temperature (*Zhu et al., 2015*; *Nugnes et al., 2015*). In addition, another plausible explanation may be the use

of different host plants (the host of lineage A is *E. camaldulensis* but in this research, it was DH201-2), which has been demonstrated in other systems (*Ferrari, Scarborough & Godfray, 2007*; *Biere & Tack, 2013*). Therefore, it is also essential to compare the differences in bacteria between *L. invasa* that parasitize different hosts.

## CONCLUSIONS

The results of this study obtained by high-throughput revealed the bacterial diversity and differences between sexes in *L. invasa*, suggesting an abundant endosymbiotic bacterial community, and some bacteria were reported in *L. invasa* for the first time. Moreover, the males harbored a more diverse bacterial community than did the females. The next research should focus on the bacteria found in this study to identify their specific ecological functions and the specific sex-based regulatory mechanism of *Rickettsia* occurrence in *L. invasa*.

## ACKNOWLEDGEMENTS

The authors thank Prof. Yongqiang He for sharing his knowledge of bacteria, the State Key Laboratory for Conservation and Utilization of Subtropical Agro-bioresources, and the members of the Guangxi Key Laboratory of Forest Ecology and Conservation.

### Funding

The research was financially supported by the National Natural Science Foundation of China (grant number 31560212, 31971664 and 31870634) and Guangxi Natural Science Foundation (grant number No. 2018GXNSFAA294008, 2018GXNSFDA281004 and 2018GXNSFAA138099). The funders had no role in study design, data collection and analysis, decision to publish, or preparation of the manuscript.

### Grant Disclosures

The following grant information was disclosed by the authors:
National Natural Science Foundation of China: 31560212, 31971664, 31870634.
Guangxi Natural Science Foundation: 2018GXNSFAA294008, 2018GXNSFDA281004, 2018GXNSFAA138099.

### Competing Interests

The authors declare there are no competing interests.

### Author Contributions

- Chunhui Guo conceived and designed the experiments, performed the experiments, analyzed the data, prepared figures and/or tables, authored or reviewed drafts of the paper, and approved the final draft.
- Xin Peng performed the experiments, authored or reviewed drafts of the paper, and approved the final draft.

- Xialin Zheng performed the experiments, analyzed the data, authored or reviewed drafts of the paper, and approved the final draft.
- Xiaoyun Wang analyzed the data, authored or reviewed drafts of the paper, and approved the final draft.
- Ruirui Wang conceived and designed the experiments, performed the experiments, prepared figures and/or tables, and approved the final draft.
- Zongyou Huang conceived and designed the experiments, prepared figures and/or tables, authored or reviewed drafts of the paper, and approved the final draft.
- Zhende Yang conceived and designed the experiments, performed the experiments, prepared figures and/or tables, authored or reviewed drafts of the paper, and approved the final draft.

### Data Availability

Data is available at NCBI SRA: SRR9591039, SRR9591038.

The COI and 16S rDNA sequences are available at GenBank: MN524231 and MN524230.

Additional data is also available at Figshare: Guo, Chunhui (2019): mic.zip. figshare. Dataset. DOI: 10.6084/m9.figshare.8323616.v1.

### Supplemental Information

Supplemental information for this article can be found online at http://dx.doi.org/10.7717/peerj.8411#supplemental-information.

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
