# Peer review of "Comparison of bacterial diversity and abundance between sexes of Leptocybe invasa Fisher & La Salle (Hymenoptera: Eulophidae) from China"

_PeerJ, doi:10.7717/peerj.8411_

## Round 0.1 · original submission · Major Revisions

Dear Dr. Guo and colleagues:

Thanks for submitting your manuscript to PeerJ. I have now received two independent reviews of your work, and as you will see, the reviewers raised some concerns about the research. Despite this, these reviewers are optimistic about your work and the potential impact it will have on research communities studying sex-specific differences in microbiota associated with Leptocybe invasa. Thus, I encourage you to revise your manuscript, accordingly, taking into account all of the concerns raised by both reviewers.

Please have an English expert review the manuscript for content, clarity and grammar. Please also address the comments in a marked-up version of your manuscript provided by Reviewer 2.

I look forward to seeing your revision, and thanks again for submitting your work to PeerJ.

Good luck with your revision,

-joe

Reviewer 1 ·

Basic reporting

• The article uses clear and unambiguous text and is conform to professional standards, but sometimes the sentences resulted to be very short and as a list of examples and respective references.
• The state of art, although brief, is enough complete.

Experimental design

• The experimental design resulted incomplete about the following several aspects:
• Although authors known the existence of different populations of L. invasa (authors cited Nugnes et al, 2015 – where the two putative species were infected by different Rickettsia bacteria) they do not characterize the studied population. This aspect could be fundamental to define the insect-symbiont system. Furthermore, Nugnes et al, 2015 speculated about the rickettsia involvement in the manipulation of the sex, but authors did not give any information about the sex-ration of their studied population.
• -In parallel, I think it’s crucial for the completeness of the manuscript that authors compare their studied population and other populations with different sex ratio (population where no males were recorded, from Italy, Tunisia, Brasil). Adding other samples with different sex-ratio could significantly make the manuscript more complete.
• In m&m paragraph, references are often not complete: in lines 118-119 there are no mention about the reference of the used primers as in lines 140-149 about used softwares.
• In results paragraph a phylogenetic tree was reported but in m&m paragraph there are no mentions about the methodology used to obtain it.

Validity of the findings

• Results paragraph should be shorter, often the same results are in the text and in the tables. I would suggest summarize the paragraph avoiding the lists of the found bacteria (order, families etc.) that are already reported in the tables, but to focus on the main bacteria.
• Taking in account that the main bacteria found in females were Rickettsia, authors should have evaluated the phylogenetic placement and the genetic diversity of these bacteria since other studies were previously carried out (Nugnes et al.,2015)
• The discussion paragraph tends to generalize too much the results found in the work. Often the authors referred to host-symbiont systems involving all the insect orders, I would suggest to be restricted to Hymenoptera order, and to refer to other orders in case no example about wasps exists.
• In the discussion paragraph, there are sometimes statements non supported by tests or reference. For example in lines 265-266: how do the authors assert the effect of bacteria in L. invasa? this statement should be supported by tests carried out with infected and uninfected strains of L. invasa.
- In lines 316-317 authors declare: “we cannot rule out the possibility that male-Rickettsia is obtained through horizontal transmission in some way”. Authors should take in account that these males represent the offspring of an infected female, hence a low titre of bacteria in males could be present because bacteria could have been transmitted from the mother itself, but the density was too low that had no effects on sex determination. Therefore, the density of the bacteria could play a crucial role in the determination of the sex, as in Giorgini et al.,2010, where Pnigalio putative males eggs obtained from cured females showed a low bacterial density, but the presence of Rickettsia was not excluded. The environimental factors could influence the density of the bacteria and hence decrease or increase their effect on the host.
Hence I would suggest to take in account the following papers:
- Jaenike J. Spontaneous emergence of a new Wolbachia phenotype. Evolution. 2007;61: 2244–2252.
- Bordenstein SR, Bordenstein SR. Temperature affects the tripartite interactions between bacteriophage WO, Wolbachia, and cytoplasmic incompatibility. PLoS ONE. 2011;6: e29106.
- Giorgini M. Induction of males in thelytokous populations of Encarsia meritoria and Encarsia protransvena: a systematic tool. BioControl. 2001;46: 427–438.

Additional comments

• Based on my previous statements, I think the work presents some gaps that need to be filled as they are essential for an appropriate study of the Leptocybe invasa bacterial pattern. Therefore, I think that this work does not deserve publication in this form and requires major revisions.

·

Basic reporting

The manuscript requires the input of a language editor to improve the grammar and sentence structure so that it is easily understandable by readers and the English used is of a professional standard.

Important literature has not been included and the information which is included on the background and context is sparse. The introduction should be expanded (as indicated in the MS) and the discussion could benefit from more reference to other literature. Most of the discussion is almost a repetition of the results section with very general statements made but not linked back to the study and highlighting the relevance of these findings to the organism of interest.

Experimental design

This study substantially elaborates on work which has been started on the genus Leptocybe, investigating bacteria present within the species. The MS could benefit from a very clearly defined research question (which will become relevant and more meaningful if more background information and context is provided - as mentioned in 1. above, this is lacking and needs substantial improvement).

A major area of concern is that the specimens used for this study have not been identified either morphologically or at a molecular level. Literature has indicated the presence of at least two haplogroups within what has been called Leptocybe invasa. Molecular confirmation of identity would be preferable as the morphological taxonomy is not fully resolved. The two haplogroups have very different global distributions and host preferences. Furthermore, the study could benefit from identifying the predominant bacteria to species level thereby agreeing with other similar literature on Leptocybe and allowing more concrete speculation (based on literature) as to the role these bacterial species could be playing in the biology of the organism.

The materials and methods section requires substantial more detail and explanation of why certain methods were chosen and implemented.

Validity of the findings

The conclusions to some extent address the results obtained but also discuss other areas which have not been addressed in the introduction or discussion in adequate detail.
For example the below sentence "These results enrich the information of microbial information of L. invasa, help research the reproductive strategy, sex control and invasive mechanism, and lay the foundation for further studies on the excavation and utilization of microbes for the biological control of L. invasa." is very broad and includes important aspects which can be further investigated and addressed with the results of this study, but no clear links have been made between the results and the potential application for control.

Additional comments

A very interesting study with interesting results. Identification of the specimens used for this study, as well as identification of the predominant bacteria to species level, would greatly increase the importance of this work. Incorporation of specimens from the two different haplogroups would improve this study even further and be of large interest and relevance to the community of individuals working with this insect pest.

Detailed comments and suggestions have been included in the manuscript, using track changes.

---

## Round 0.2 · Major Revisions

Dear Dr. Guo and colleagues:

Thanks for re-submitting your manuscript to PeerJ. I have only received a re-review from reviewer 1, but it appears that you have not addresses most of the concerns by both of the original reviewers.

Accordingly, please revisit the original reviews of your work and revise your manuscript. It is unacceptable to gloss over the criticisms raised by the reviewers.

Please let me know if you have any problems with this decision.

Best,

-joe

Reviewer 1 ·

Basic reporting

Authors improved the manuscript, taking in account the comments suggested by the second referee directly in the pdf of the first version of the manuscript. However, some important information in the introduction and discussion should be treated better, sometimes in the discussions some results are not sufficiently treated in connection with L. invasa biology or management.

Experimental design

The experimental design shows the same problems highlighted by referees in their previous reviews. Authors did not take in account all the suggestions made by the referees, but they only based on the comments reported directly on the manuscript by the second referee. Thus, the manuscript has improved as regard the form and the grammar, but the main themes still show lacking argumentation and development. Furthermore, although authors added some information about the L. invasa characterization, they completely overlooked the methods they used and, in my opinion, obtained sequences should be published on sequence database. The sex ratio was not calculated on the studied population, but referred to a previous work (Zheng et al., 2018). Based on the literature about endosymbiotic bacteria, the titre of bacteria (especially those manipulating the reproduction) could change due to environmental characteristics (temperature) that could lead a difference in the sex ratio.

Validity of the findings

In this new version of the manuscript, results did not change. Authors try to streamline the paragraph avoiding the list of found bacteria but, on the other side, authors did not identify the predominant bacteria to species level as previously request by the two referees.
I can not find any crucial change respect to the previous version of the manuscript.
Furthermore, the last part of the conclusions is completely confused about the use of sterile males in a partially thelytokous population (where males are not crucial for the reproduction) and, about the use of antibiotics in the change of offspring.

Additional comments

As already mentioned, the authors modified the manuscript only based on comments on pdf. They completely did not consider the main suggestions by the two referees to improve their research.

Therefore, I think that this work does not deserve publication in this form and requires again major revisions.

---

## Round 0.3 · Minor Revisions

Dear Dr. Guo and colleagues:

Thanks for re-submitting your manuscript to PeerJ. Another re-review from reviewer 1 indicates much improvement, though there are some concerns with English and grammar.

Accordingly, please address these concerns and also enlist the help of an English expert on your revision.

Once these issues are addressed we may move towards acceptance of your work.

Best,

-joe

Reviewer 1 ·

Basic reporting

The manuscript needs a language editor update, some sentences are not clear. It could improve the whole MS.

Experimental design

Most of aspects were improved by authors.It is a pity authors could not find other populations (from other countries - or other lineage) to compare them. Please see my suggestions at the end.

Validity of the findings

Results and Discussion were clearly summarized and improved.
Please read my general comments.

Additional comments

Authors followed the most of the suggestions of the reviewers. Nevertheless I need to make some considerations:
- in the first revision it was suggested to compare the Chinese L.invasa with those of other countries, I imagine that the authors failed to obtain these samples. It is therefore important to stress, already from the title, that the research is focused on L. invaded from China. I suggest to change the title.
- In my opinion, the phylogenetic reconstructions trees without outgroup, without the most important RIckettisia species, without the information in M&m paragraph (i.e. number of bootstrap or generations) is not befitting fo PeerJ journal. I would suggest the authors to make a simple blast against genebank database and take in account the similarity with lineages (for L. invasa) and rickettsia group.
- molecular characterization: the PCR program refers to a single genes or both of them? (COI-16S)
- line 99: “…fast growth of L. invasa” please add populations.
- line 128. The sentence “In this research, a total of 656 females and 51 males were collected (Table S1). “ should be in “results” paragraph.
- 224: transitional family: please change in transitional group.
- from line 332 to 336: please rephrase, some sentences are not clear.

In conclusion paragraph:
- “bacterial endosymbionts” please change with “endosymbiotic bacteria”;
- “Some bacterial” please change with “some bacteria”
- i think 340-342 is ripetitive period. The same concept with different sentences. Please rephrase

---

## Round 0.4 · accepted · Accept

Dear Dr. Guo and colleagues:

Thanks for re-submitting your revised manuscript to PeerJ, and for addressing the concerns raised by the reviewer. I now believe that your manuscript is suitable for publication. Congratulations! I look forward to seeing this work in print, and I anticipate it being an important resource for research communities studying sex-specific differences in microbiota associated with Leptocybe invasa.

Thanks again for choosing PeerJ to publish such important work.

-joe